# Mining Multi-Label Samples from Single Positive Labels

**Youngin Cho*[1]**  **Daejin Kim*[1,2]**  **Mohammad Azam Khan[1]**  **Jaegul Choo[1]**

[1]KAIST AI   [2]NAVER WEBTOON AI

{choyi0521,kiddj,azamkhan,jchoo}@kaist.ac.kr

## Abstract

Conditional generative adversarial networks (cGANs) have shown superior results in class-conditional generation tasks. To simultaneously control multiple conditions, cGANs require multi-label training datasets, where multiple labels can be assigned to each data instance. Nevertheless, the tremendous annotation cost limits the accessibility of multi-label datasets in real-world scenarios. Therefore, in this study we explore the practical setting called the *single positive setting*, where each data instance is annotated by only one positive label with no explicit negative labels. To generate multi-label data in the single positive setting, we propose a novel sampling approach called single-to-multi-label (S2M) sampling, based on the Markov chain Monte Carlo method. As a widely applicable "add-on" method, our proposed S2M sampling method enables existing unconditional and conditional GANs to draw high-quality multi-label data with a minimal annotation cost. Extensive experiments on real image datasets verify the effectiveness and correctness of our method, even when compared to a model trained with fully annotated datasets.

## 1 Introduction

Since being proposed by Goodfellow *et al.* [1], generative adversarial networks (GANs) have gained much attention due to their realistic output in a wide range of applications, *e.g.*, image synthesis [2, 3, 4], image translation [5, 6, 7, 8], and data augmentation [9, 10]. As an advanced task, generating images from a given condition has been achieved by conditional GANs (cGANs) and their variants [11, 12]. To reflect the nature of real data where each data instance can belong to multiple classes, multi-label datasets have been introduced in the applications of cGANs [8, 13]. In a multi-label dataset, each data instance can be specified with multiple attributes. For example, in a multi-label facial image dataset, each image is labeled for entire classes such as *Black-hair*, *Smile*, and *Male*. The label for each class is given as a binary value and indicates the presence or absence of the corresponding attribute. Despite its usefulness, as claimed in earlier studies [14, 15, 16], access to multi-label datasets is severely limited in practice due to the difficulty of annotating all attributes. Under these circumstances, a weakly annotated dataset is used as an alternative for cGANs to reduce the annotation cost.

In this paper, we introduce the *single positive setting* [16], originally proposed for classification tasks, to conditional generation. Each data instance has a label indicating only the presence of one attribute (*i.e.*, a single positive label) in this setting, and the presence of the rest of the attributes remains unknown. For instance, in a facial image dataset, all attributes are fully specified in a multi-label setting (*e.g.*, *Smiling black-haired man*) whereas only one attribute is specified in the single positive setting (*e.g.*, *Black-hair*). The single positive setting allows us not only to reduce the annotation cost, but also to model the intrinsic relationships among classes.

---

*Equal contribution

36th Conference on Neural Information Processing Systems (NeurIPS 2022).

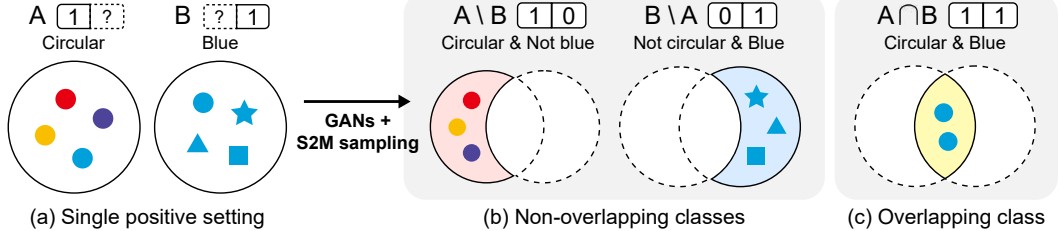

Figure 1: Illustration of joint classes where two attributes (circular, blue) are given. (a) Only one of the attributes is specified in the single positive setting. From classes $A$ and $B$, our proposed S2M sampling method can draw samples from three joint classes: (b) two non-overlapping classes ($A \setminus B$ and $B \setminus A$) and (c) one overlapping class ($A \cap B$).

To generate multi-label data using only single positive labels, we consider two types of combinatorial classes from two classes, $A$ and $B$, each corresponding to a single positive label: (1) *overlapping class*, where the data instances belong to both classes ($A \cap B$), and (2) *non-overlapping class*, where the data instances belong to only one of the classes ($A \setminus B$ or $B \setminus A$). Figure 1 shows examples of overlapping class and non-overlapping classes when two types of single positive labels are given. By extending this concept, we can consider combinatorial classes, where data instances belong to certain classes but not to the rest. We denote these classes as *joint classes*. By accessing a joint class, we can generate multi-label samples using only single positive labels. Ideally, we can represent all possible label combinations with at least one positive label.

Several attempts have been made to consider such a joint class in generation tasks. Specially designed generative models such as GenPU [17] and RumiGAN [18] were proposed to generate samples from one class while excluding another class. However, these studies deal with only two classes without considering the overlapping class. Recently, Kaneko *et al.* [19] proposed CP-GAN to capture between-class relationships in conditional generation. To generate images belonging to $n$ classes, they equally assign $1/n$ as the class posterior for each selected class. This indicates that CP-GAN generates images that have an equal probability of being classified as each class. However, because an image in the real world that belongs to multiple classes does not have equal class posteriors, sample diversity is lost. Consequently, we focus on a sampling approach to precisely draw samples from complex distributions that are difficult to directly sample. In recent GANs studies, sampling approaches [20, 21] employed the rejection sampling or Markov chain Monte Carlo method to obtain realistic data. These approaches adopt the sampling process as the post-processing method of GANs and filter generated images using the pretrained classifiers.

In line with these studies, we propose a novel sampling framework called *single-to-multi-label (S2M) sampling* to correctly generate data from both overlapping and non-overlapping classes. We newly propose a tractable formulation to estimate the conditional density of joint classes. Concretely, we employ several classification networks to estimate the target conditional density and apply the Markov chain Monte Carlo method to pretrained unconditional and conditional GANs. In Figure 2, we depict the conceptual difference against the existing approaches (1st row) and provide the empirical results on a 1D Gaussian example (2nd row). As S2M sampling performs at the inference time of generation, it fully preserves the image quality and eliminates the need for changing the objective functions and architectures. We validate the effectiveness of our method compared to the existing models on diverse datasets such as MNIST, CIFAR-10 and CelebA. To the best of our knowledge, our approach is the first sampling framework that generates multi-label data from single positive labels. Our contributions can be summarized as follows:

- We introduce the single positive setting in the conditional generation task and provide the theoretical framework for estimating conditional densities of joint classes.

- We propose a novel sampling framework based on the Markov chain Monte Carlo method for generating multi-label data from single positive labels.

- Through extensive experiments, we show that our proposed S2M sampling method correctly draws multi-label data while fully preserving the quality of generated images.

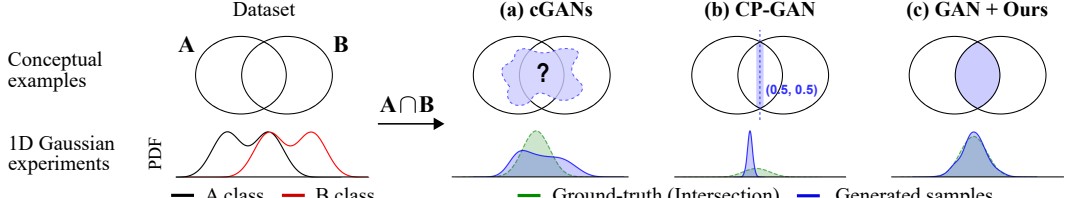

Figure 2: cGANs, CP-GAN and our method are compared in a class-overlapping case. 1D Gaussian examples consists of two classes of one-dimensional Gaussian mixtures with one common mode, and each method attempts to generate samples in the overlapping region. For cGANs and CP-GAN, an equal value of $0.5$ is provided as labels for the two classes. (a) It is not clear how cGANs obtain samples of the class. (b) CP-GAN draws samples from the narrow region. (c) GAN with S2M sampling draws samples accurately without sacrificing diversity.

## 2    Related Work

**Conditional GANs.** The aim of conditional GANs (cGANs) [11] is to model complex distributions and control data generation by reflecting the label input. Various studies of cGANs have made significant advances in class-conditional image generation by introducing an auxiliary classifier [12, 22], modifying the architecture [23, 2], and applying metric learning [24]. In a weakly-supervised setting, GenPU [17] and RumiLSGAN [18] specify only two classes and draw samples that belong to one class but not the other. CP-GAN [19] learns to draw samples conditioned on the probability output of the classifier. Given that this model tends to draw samples on a limited region of the data space, it is challenging to ensure a variety of samples as shown in Figure 2. In contrast, we propose a sampling method that draws multi-label samples without sacrificing diversity.

**Sampling in GANs.** Sampling methods are used to improve the sample quality in GANs. Discriminator rejection sampling [20] uses the scheme of rejection sampling and takes samples close to real data by estimating the density ratio with the discriminator. In addition, Metropolis-Hastings GAN [21] adopts the Markov chain Monte Carlo (MCMC) method and calibrates the discriminator to improve the sample quality in a high-dimensional data space. Discriminator driven latent sampling [25] uses the MCMC method in the latent space of GANs to draw realistic samples efficiently. GOLD estimator [26] uses a sampling algorithm to improve the quality of images for class-conditional generation. While previous studies focus on improving the sample quality, our S2M sampling method aims to draw multi-label samples while also improving sample quality.

## 3    Methods

### 3.1    Problem Setting

Let $x \in X$ be a data point as a random variable and let $y_{1:n} \in \{0, 1\}^n$ denote its corresponding multi-labels as binary random variables. Here, for every $k$, $y_k = 1$ indicates that $x$ is contained in the $k$-th class while $y_k = 0$ indicates that $x$ is not. We consider two index sets, an intersection index set $I$ and a difference index set $J$, so that the pair $(I, J)$ can be used as an index to indicate the class, where data points contained in all classes indicated by $I$ but excluded from all classes indicated by $J$. Let $\mathcal{I}$ be a collection of all possible pairs of $I$ and $J$, defined as

$$\mathcal{I} = \{(I, J) \in \mathcal{P}(N) \times \mathcal{P}(N) : I \neq \emptyset, I \cap J = \emptyset\}, \tag{1}$$

where $N = \{1, 2, ..., n\}$ is a finite index set of all classes and $\mathcal{P}(N)$ is the power set of $N$. That is, the intersection index set indicates at least one class and is distinct from the difference index set. Especially for $I \cup J = N$, the class indicated by $(I, J)$ is called the *joint class*.

Let $p(x, y_1, y_2, ..., y_n)$ be the joint probability density function, and let $p_{data}(x, c)$ be the joint density of an observed data point $x \in X$ and a class label $c \in N$ such that $p_{data}(x|c) = p(x|y_c = 1)$. Given the class priors $p_{data}(c), \pi_c := p(y_c = 1)$ and samples drawn from the class-conditional density $p_{data}(x|c)$ for each $c = 1, 2, ..., n$, our goal is to draw samples from the conditional density

$$p_{(I,J)}(x) := p(x|\forall i \in I, \forall j \in J, y_i = 1, y_j = 0), \tag{2}$$

for $(I, J) \in \mathcal{I}$ and $\pi_{(I,J)} := p(\forall i \in I, \forall j \in J, y_i = 1, y_j = 0) > 0$. In this work, we propose adding a mild constraint which will allow our sampling algorithm to derive the target density.

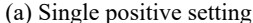
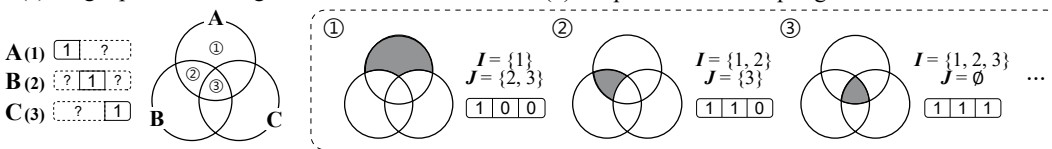

Figure 3: (a) A dataset with single positive labels is given. (b) S2M sampling can draw samples as multi-label data with two index sets: intersection ($I$) and difference ($J$).

**Assumption 1.** *For every $i, j \in N$ such that $i \neq j$, if $p(y_i = 1, y_j = 0) > 0$ and $p(y_j = 1, y_i = 0) > 0$, then $\mathrm{supp}\, p(x|y_i = 1, y_j = 0) \cap \mathrm{supp}\, p(x|y_j = 1, y_i = 0) = \emptyset$.*

Assumption 1 states that no data points are likely to be assigned two mutually exclusive labels, which can be naturally assumed in many practical situations. Figure 3 illustrates our problem setting.

### 3.2 Mining Multi-Label Data with S2M Sampling

Naturally, a question may arise as to whether the supervision given to us is sufficient to obtain multi-label samples. To gain insight into this, we derive a useful theorem which provides an alternative formulation for the target density (2).

**Theorem 1.** *Let $\{f_{(I,J)} : X \to [0, \infty)\}_{(I,J) \in \mathcal{I}}$ be an indexed family of non-negative measurable functions on $X$, and let $f_k := f_{(\{k\}, \emptyset)}$. Then, the following conditions hold:*

*(a) $\forall (I, J) \in \mathcal{I}, f_{(I,J)} = \sum_{S: I \subseteq S, J \subseteq N \setminus S} f_{(S, N \setminus S)}$*
*(b) $\forall i, j \in N \text{ s.t. } i \neq j, \mathrm{supp}\, f_{(\{i\}, \{j\})} \cap \mathrm{supp}\, f_{(\{j\}, \{i\})} = \emptyset$*

*if and only if, for every $(I, J) \in \mathcal{I}$,*

$$f_{(I,J)} = \begin{cases} (\min_{i \in I} f_i - \max_{j \in J} f_j)^+ & \text{if } J \neq \emptyset \\ \min_{i \in I} f_i & \text{otherwise} \end{cases}, \tag{3}$$

*where $(\cdot)^+$ represents the positive part.*

*Proof.* Please refer to Appendix A. $\square$

Let $f_{(I,J)}(x) = p(x, \forall i \in I, \forall j \in J, y_i = 1, y_j = 0)$ for every $(I, J) \in \mathcal{I}$. Then, both $(a)$ and $(b)$ in Theorem 1 hold by the Assumption 1. According to the Theorem 1, if $\pi_{(I,J)} > 0$,

$$p_{(I,J)}(x) = \pi_{(I,J)}^{-1}(\min\{\pi_i p(x|y_i = 1) : i \in I\} - \max\{\pi_j p(x|y_j = 1) : j \in J\} \cup \{0\})^+. \tag{4}$$

The alternative formula (4) shows that the density of the joint class can be derived from the class-conditional densities of the single positive labels. Despite their clear relationship, training a generator that can model the density $p_{(I,J)}$ is a non-trivial problem since the formula consists of several implicitly defined conditional densities, class priors, and variable sets. To address this issue, we propose the application of sampling approaches upon existing GANs. Interestingly, our sampling framework called S2M sampling allows not only for samples to be drawn from the target density, but also the modification of $I$, $J$, and class priors at inference time. The rest of this section introduces the main approaches of our S2M sampling method.

**Density Ratio Estimation.** We utilize several classification networks to compute implicitly defined density ratios. For simplicity, we denote $G$ as a pretrained generator for both unconditional and conditional GANs. $G$ produces data $x$ by taking a latent $z$ and a class label $c$ for class-conditional generation and only $z$ for unconditional generation. We consider three classifiers $D_v, D_r$, and $D_f$ which are obtained by minimizing $\mathcal{L}_v, \mathcal{L}_r$, and $\mathcal{L}_f$, respectively, *i.e.*,

$$\begin{aligned} \mathcal{L}_v &= -\,\mathbb{E}_{(x,c) \sim p_{data}(x,c)}[\log D_v(x)] - \mathbb{E}_{x \sim p_G(x)}[\log(1 - D_v(x))], \\ \mathcal{L}_r &= -\,\mathbb{E}_{(x,c) \sim p_{data}(x,c)}[\log D_r(c|x)], \; \mathcal{L}_f = -\mathbb{E}_{(x,c) \sim p_G(x,c)}[\log D_f(c|x)], \end{aligned} \tag{5}$$

where $p_G$ is the distribution of the generated samples by $G$. The optimal classifiers trained by these losses $D_v^*, D_r^*$, and $D_f^*$ satisfy the following equations: $D_v^*(x) = p_{data}(x)/(p_{data}(x) + p_G(x)), D_r^*(c|x) = p(x|y_c = 1)p_{data}(c)/p_{data}(x), D_f^*(c|x) = p_G(x|c)p_G(c)/p_G(x)$. From $D_v^*$,

$D_r^*$, and $D_f^*$, we can access the density ratios $p_{data}(x)/p_G(x)$, $p(x|y_c = 1)/p_{data}(x)$, and $p_G(x|c)/p_G(x)$ which will be used to compute the acceptance probability of the MCMC method.

**S2M Sampling for Unconditional GANs.** We apply Metropolis-Hastings (MH) independence sampling [27, 21] to draw samples from the complex target distribution $p_t$. The MH algorithm uses a Markov process where each transition from a current state $x_k$ to the next state $x_{k+1}$ is made by an accept-reject step. At each step of MH independent sampling, a new proposal $x'$ is sampled from a proposal distribution $q(x'|x) = q(x')$ and is then accepted with probability $\alpha(x', x)$ which is given by $\alpha(x', x) = \min\left(1, \frac{p_t(x')q(x)}{p_t(x)q(x')}\right)$. We set $x_{k+1} = x'$ if $x'$ is accepted, and $x_{k+1} = x_k$ otherwise. Under mild conditions, the chain $x_{1:K}$ converges to the unique stationary distribution $p_t$ as $K \to \infty$[1]

Let $G$ be a generator where the support of $p_G$ contains that of $p_{(I,J)}$. To draw samples from $p_{(I,J)}$, we let $p_t = p_{(I,J)}$ and take multiple samples from $G$ as independent proposals, i.e. $q(x'|x) = p_G(x')$. Then, the desired acceptance probability $\alpha(x', x)$ can be calculated using $D_v^*$ and $D_r^*$:

$$
r_{(I,J)}(x) := \left(\min\left\{\frac{\pi_i}{p_{data}(i)}D_r^*(i|x) : i \in I\right\} - \max\left\{\frac{\pi_j}{p_{data}(j)}D_r^*(j|x) : j \in J\right\} \cup \{0\}\right)^+,
$$

$$
\alpha(x', x) = \min\left(1, \frac{p_{(I,J)}(x')/p_G(x')}{p_{(I,J)}(x)/p_G(x)}\right) = \min\left(1, \frac{r_{(I,J)}(x')(D_v^*(x)^{-1} - 1)}{r_{(I,J)}(x)(D_v^*(x')^{-1} - 1)}\right).
$$

(6)

Different from Turner *et al.* [21], the term $r_{(I,J)}(x')/r_{(I,J)}(x)$ is added to the acceptance probability formula to draw multi-label samples. To obtain uncorrelated samples, a sample is taken after a fixed number of iterations for each chain. The sampling approach allows one to control the parameters $I$, $J$, and $\gamma_k := \pi_k/p_{data}(k)$ without any additional training of the model. A summary of our S2M sampling algorithm is provided in Appendix B.1.

**S2M Sampling for Conditional GANs.** cGANs can provide a proposal distribution close to the target distribution $p_{(I,J)}$, which greatly increases the sample efficiency of the MCMC method. Let $c$ be a class label such that the support of class-conditional density $p_G(\cdot|c)$ contains that of $p_{(I,J)}$. At each step of the MH algorithm, the proposal $x' \sim q(x'|x) = p_G(x'|c)$ is accepted with a probability $\alpha_c(x', x)$. The desired $\alpha_c(x', x)$ can be calculated as

$$
\alpha_c(x', x) = \min\left(1, \frac{p_{(I,J)}(x')/p_G(x'|c)}{p_{(I,J)}(x)/p_G(x|c)}\right) = \min\left(1, \frac{r_{(I,J)}(x')D_f^*(c|x)(D_v^*(x)^{-1} - 1)}{r_{(I,J)}(x)D_f^*(c|x')(D_v^*(x')^{-1} - 1)}\right).
$$

(7)

That is, the sampling method can be adopted to cGANs by additionally computing $D_f^*(c|x)/D_f^*(c|x')$.

**Latent Adaptation in S2M Sampling.** While our S2M sampling method allows us to draw multi-label samples, the algorithm rejects certain samples through the sampling procedure if the target distribution $p_{(I,J)}$ is significantly different from the generator distribution. Specifically, an independent MH algorithm takes time inversely proportional to the probability of a generated sample belonging to the target class. To alleviate this issue, we propose a technique to improve the sample efficiency from past sampling attempts. Initially, a certain number of pilot target class samples $x_{1:m}$ are drawn using S2M sampling, and then the corresponding target latent samples $t_{1:m}$; $x_k = G(t_k)$ are obtained. Since the latent samples are nearly restricted to the latent prior of GANs (*e.g.*, Gaussian distribution), the distribution $\tilde{p}_z$ of the target latent can be roughly estimated by fitting a simple probabilistic model using $t_{1:m}$. Let $\tilde{p}_G$ be the newly obtained generator distribution induced by $G$, *i.e.*, $G(z) = x \sim \tilde{p}_G(x)$ for $z \sim \tilde{p}_z(z)$. The sample efficiency can be further improved by using $\tilde{p}_G$ as the proposal distribution in the MH algorithm since $\tilde{p}_G$ is much close to the target distribution. To run the MH algorithm without retraining the classifiers, we approximate $p_G(G(z))/\tilde{p}_G(G(z)) \approx C \cdot p_z(z)/\tilde{p}_z(z)$[2] for some constant $C$. Then, at each step of the MH algorithm, the proposal $x' \sim q(x'|x) = \tilde{p}_G(x')$ is accepted

---

[1]For example, the chain of samples is uniformly ergodic if $p_t/q$ is bounded [28].

[2]The equation is derived in the previous latent sampling approaches by the reverse lemma of rejection sampling [25] or the change-of-variables formula [29].

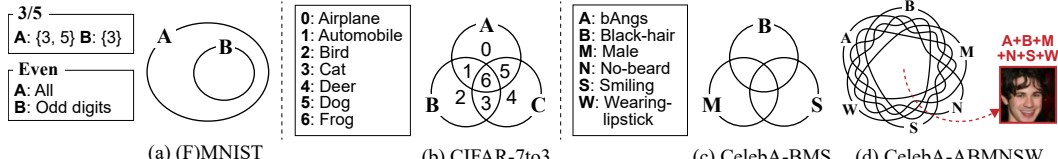

Figure 4: Experimental settings for (F)MNIST, CIFAR-7to3, CelebA-BMS, and CelebA-ABMNSW.

with a probability $\tilde{\alpha}(x', x)$ which can be calculated as

$$\tilde{\alpha}(x', x) = \min\left(1, \frac{p_{(I,J)}(x')/\tilde{p}_G(x')}{p_{(I,J)}(x)/\tilde{p}_G(x)}\right) \approx \min\left(1, \frac{r_{(I,J)}(x')(D_v^*(x)^{-1} - 1)p_z(z')/\tilde{p}_z(z')}{r_{(I,J)}(x)(D_v^*(x')^{-1} - 1)p_z(z)/\tilde{p}_z(z)}\right),$$
(8)

where $x' = G(z')$ and $x = G(z)$. Here, we can explicitly compute the density ratio between $p_z$ and $\tilde{p}_z$ by letting $\tilde{p}_z$ be a known probability density function such as a Gaussian mixture. In practice, if the target class samples rarely appear in the generated samples, one can consider applying latent adaptation repeatedly. For instance, to draw *Black-haired man* samples, we can first search the latent space of *Man* by using latent adaptation, and then perform it in that space again to search the space of *Black-haired man*. In terms of time complexity, this is performed more efficiently than searching the space of *Black-haired man* at once. A detailed description of latent adaptation and a discussion about the time complexity are provided in Appendix B.2 and Appendix C, respectively.

**Practical Considerations.** We employ three classifiers $D_v$, $D_r$, and $D_f$, to compute the acceptance probability used in the MCMC method. For better training efficiency of the classifiers, we use shared layers, except for the last linear layer. Since the classifiers are not optimal in practice, we scale the temperature of classifiers [30] and adjust $\gamma_k$ to calibrate the sampling algorithm. Detailed settings and the ablation study are provided in Appendix D and Appendix E.4, respectively.

## 4 Experiments

In this section, we validate that our S2M sampling method properly draws samples within the joint classes. Specifically, we mainly consider two cases in the single positive setting: (i) one class is entirely contained in another class, (ii) multiple classes are partially overlapping with each other. In the former setting, we verify that S2M sampling can completely filter out samples of a smaller class, and compare our method to the existing models including GenPU [17] and RumiLSGAN [18] on MNIST and Fashion-MNIST (FMNIST). In the latter setting, all possible joint classes are assessed with our S2M sampling method. Subsequently, we evaluate how accurately S2M sampling draws samples in these classes compared to CP-GAN [19] and the models trained with a fully annotated multi-label dataset.

To verify whether a method can accurately generate samples for each joint class, we evaluate *accuracy* which indicates how many generated images are assigned to the target joint classes by a classifier trained with fully annotated data (*i.e.*, multi-label data). Since accuracy itself cannot fully evaluate the distance between the distribution of generated samples and that of real samples, we additionally introduce various metrics to evaluate fidelity (*i.e.*, how realistic the generated samples are) and diversity of generated images: (i) *Inception Score* (IS) [31], (ii) *Fréchet Inception Distance* (FID) [32], (iii) *improved precision and recall* [33], and (iv) *density and coverage* [34].

FID and IS are metrics to evaluate the fidelity and the diversity of generated samples using the features of the pretrained Inception-V3 network [35]. A lower FID and a higher IS indicate higher fidelity and diversity of generated samples. Precision and density measure the ratio of generated samples that falls within the manifold of real samples, so that they evaluate the fidelity of generated samples. Contraily, recall and coverage measure the ratio of real samples that falls within the manifold of the generated samples, and thus evaluating their diversity. That is, a higher value of these metrics indicates that the generated samples have a distribution similar to the real one.

Since several metrics for fidelity and diversity are not applicable for non-ImageNet-like images (*e.g.*, MNIST and FMNIST), we instead use the FID score computed from activations of LeNet5 [36] and denote this as FID$^\dagger$. The test set is used as a reference dataset for evaluating FID. All quantitative results are averaged over three independent trials, and the standard deviation is denoted by subscripts.

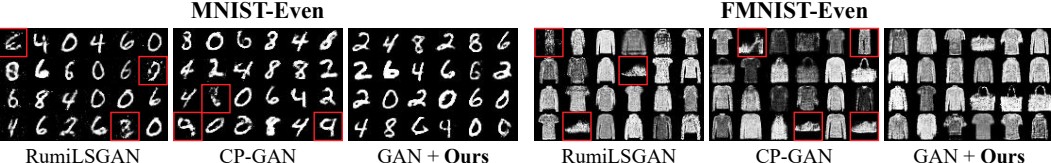

Figure 5: Qualitative results on MNIST-Even and FMNIST-Even.

The qualitative results are randomly sampled in all experiments. Detailed experimental settings, such as architectures and hyperparameters, are provided in Appendix D.

## 4.1 Sampling for Classes with Inclusion

**Experimental Settings.** We consider a special case of our problem setting, where one class is entirely contained in another class ($B \subset A$) as shown in Figure 4 (a). This setting is similar to the positive-unlabeled setting [37, 38] where the positive data is included in the unlabeled data. Under this constraint, GenPU [17], RumiLSGAN [18], and CP-GAN [19] can be used to generate samples from the non-overlapping class ($A \setminus B$). Along with these models, we attempt to draw samples from the non-overlapping class by adopting S2M sampling to unconditional WGAN-GP [39]. These experiments were conducted in three settings (See Figure 4): (i) **MNIST-3/5** ($A=\{3, 5\}$, $B=\{3\}$), (ii) **MNIST-Even** ($A=\{0, 1, 2, 3, 4, 5, 6, 7, 8, 9\}$, $B=\{1, 3, 5, 7, 9\}$), and (iii) **FMNIST-Even** ($A=\{0_{\text{T-shirt/Top}}, 1_{\text{Trouser}}, 2_{\text{Pullover}}, 3_{\text{Dress}}, 4_{\text{Coat}}, 5_{\text{Sandal}}, 6_{\text{Shirt}}, 7_{\text{Sneaker}}, 8_{\text{Bag}}, 9_{\text{Ankle boot}}\}$, $B=\{1, 3, 5, 7, 9\}$). For S2M sampling, samples are obtained at 100 MCMC iterations.

**Quantitative Results.** Table 1 shows the results of our S2M sampling method and the baselines for the non-overlapping classes. The performance of GenPU is reported only for MNIST-3/5 due to its mode collapse issue [40, 41]. S2M sampling adopted to unconditional GAN shows promising results in terms of both accuracy and FID[†]. In fact, our results are superior to the existing methods specially designed for generating non-overlapping data such as GenPU and RumiLSGAN. Notably, S2M sampling improves accuracy by 8.5% and 6.8% while decreasing FID[†] compared to the second-best models on MNIST-Even and FMNIST-Even, respectively. This indicates that S2M sampling accurately samples the images of the non-overlapping classes without being biased to a specific mode.

Table 1: Results for different models on MNIST and FMNIST.

| Method | MNIST-3/5 | | MNIST-Even | | FMNIST-Even | |
|---|---|---|---|---|---|---|
| | Acc. (%) (↑) | FID[†] (↓) | Acc. (%) (↑) | FID[†] (↓) | Acc. (%) (↑) | FID[†] (↓) |
| GenPU [17] | 99.33±0.56 | 1.93±1.10 | - | - | - | - |
| RumiLSGAN [18] | 77.06±1.54 | 13.20±1.19 | 86.11±4.83 | 3.44±1.39 | 91.07±0.88 | 3.23±2.48 |
| CP-GAN [19] | 66.89±1.46 | 19.50±0.90 | 87.87±0.40 | 2.23±0.08 | 81.21±0.67 | 6.14±0.56 |
| **GAN + Ours** | **99.52±0.34** | **0.88±0.21** | **96.37±0.25** | **0.86±0.24** | **97.87±0.66** | **2.48±0.29** |

**Qualitative Results.** Figure 5 shows the qualitative results of RumiLSGAN, CPGAN and our method adopted to WGAN-GP. As indicated by the red boxes, RumiLSGAN and CP-GAN fail to completely eliminate the samples of the smaller class (*e.g.*, odd digits for MNIST-Even or *Sneaker* for FMNIST-Even). Conversely, S2M sampling correctly draws samples for the target classes.

## 4.2 Sampling for Multiple Classes

**Experimental Settings.** Here, we consider a general case of single positive setting where multiple classes can be overlapping as shown in Figure 4 (b-d). Given $n$ classes, at most $2^n - 1$ joint classes can be obtained. In this setting, we attempt to obtain samples of these joint classes using only single positive labels. We consider cGANs with a projection discriminator (cGAN-PD) [23], AC-GAN [12], and CP-GAN [19] as the baseline generative models.

Since traditional conditional GANs such as cGAN-PD and AC-GAN are not originally designed for generating joint classes, we naively introduce these models in our setting with slight modifications. Concretely, as a method to generate images belonging to $n$ classes, we provide $1/n$ as the conditional value for each class, following Kaneko *et al.* [19]. In our experiments, we expect these models to have a lower bound performance of our results and indicate their results with asterisks (*). In contrast, conditional GANs trained on a fully annotated dataset where all joint classes are specified can be considered as strong baselines in our settings. cGAN-PD and AC-GAN are trained in this setting,

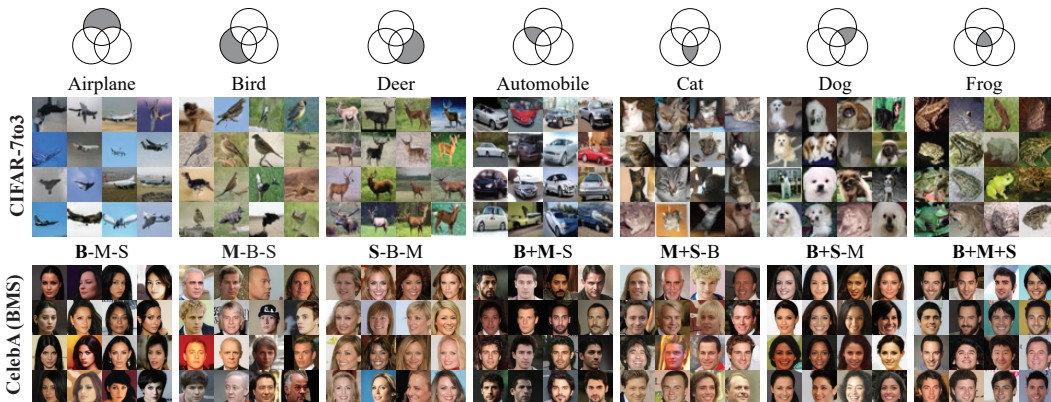

Figure 6: Results of S2M sampling with cGAN-PD on CIFAR-7to3 and CelebA-BMS. The first row depicts the target joint class.

Table 2: Accuracy (%), FID, IS, precision (P), recall (R), density (D), and coverage (C) for different models on real-world datasets.

| | Model | Acc. (↑) | FID (↓) | IS (↑) | P (↑) | R (↑) | D (↑) | C (↑) |
|---|---|---|---|---|---|---|---|---|
| CIFAR-7to3 | cGAN-PD (Oracle) [23] | 87.60±0.15 | 16.55±0.69 | 8.39±0.18 | 0.73±0.01 | 0.65±0.01 | 0.89±0.05 | 0.85±0.02 |
| | AC-GAN (Oracle) [12] | 94.14±0.39 | 16.14±0.43 | 8.44±0.03 | 0.74±0.00 | 0.63±0.00 | 0.89±0.02 | 0.84±0.00 |
| | cGAN-PD* [23] | 27.19±0.53 | 21.12±0.62 | 7.94±0.02 | 0.71±0.01 | 0.66±0.00 | 0.81±0.03 | 0.78±0.01 |
| | AC-GAN* [12] | 31.06±1.17 | 26.42±0.82 | 7.31±0.10 | 0.69±0.01 | 0.66±0.01 | 0.71±0.01 | 0.70±0.01 |
| | CP-GAN [19] | 42.62±0.54 | 27.88±2.00 | 7.42±0.06 | 0.71±0.00 | 0.66±0.00 | 0.76±0.00 | 0.71±0.02 |
| | **GAN + Ours** | 77.65±1.22 | 14.42±0.55 | 8.71±0.16 | 0.72±0.00 | **0.67±0.00** | 0.92±0.02 | 0.87±0.01 |
| | **cGAN-PD + Ours** | **80.62±2.08** | **14.14±0.34** | 9.06±0.18 | 0.75±0.01 | 0.66±0.01 | 1.09±0.12 | 0.90±0.01 |
| CelebA-BMS | cGAN-PD (Oracle) [23] | 78.35±1.99 | 10.38±0.55 | 2.45±0.07 | 0.79±0.03 | 0.48±0.02 | 1.12±0.18 | 0.86±0.02 |
| | AC-GAN (Oracle) [12] | 91.57±0.43 | 10.41±1.52 | 2.40±0.03 | 0.79±0.02 | 0.49±0.04 | 1.18±0.12 | 0.86±0.03 |
| | cGAN-PD* [23] | 39.84±0.70 | 10.97±1.56 | 2.32±0.06 | 0.79±0.01 | 0.48±0.04 | 1.19±0.05 | 0.86±0.02 |
| | AC-GAN* [12] | 52.01±1.81 | 12.39±1.02 | 2.32±0.10 | 0.79±0.02 | 0.49±0.01 | 1.17±0.09 | 0.85±0.02 |
| | CP-GAN [19] | 87.36±2.05 | 13.38±1.36 | 2.43±0.09 | 0.78±0.01 | 0.45±0.02 | 1.08±0.04 | 0.82±0.02 |
| | **GAN + Ours** | 85.22±4.27 | **10.50±0.97** | **2.51±0.08** | 0.83±0.02 | 0.48±0.03 | 1.36±0.08 | **0.89±0.02** |
| | **cGAN-PD + Ours** | **90.44±1.05** | 10.63±0.29 | 2.46±0.06 | **0.84±0.00** | 0.48±0.02 | 1.40±0.02 | 0.88±0.01 |
| CelebA-ABMNSW | cGAN-PD (Oracle) [23] | 69.61±2.91 | 13.54±2.54 | 2.44±0.11 | 0.79±0.02 | 0.42±0.04 | 1.15±0.10 | 0.84±0.02 |
| | AC-GAN (Oracle) [12] | 91.37±1.14 | 13.46±1.29 | 2.30±0.05 | 0.80±0.02 | 0.36±0.01 | 1.17±0.09 | 0.82±0.01 |
| | cGAN-PD* [23] | 15.28±0.08 | 11.09±0.67 | 2.32±0.03 | 0.80±0.01 | **0.47±0.03** | 1.19±0.07 | 0.85±0.02 |
| | AC-GAN* [12] | 23.97±0.64 | 11.92±2.06 | 2.44±0.06 | 0.77±0.04 | 0.45±0.02 | 1.11±0.13 | 0.82±0.04 |
| | CP-GAN [19] | 79.09±1.14 | 14.97±4.18 | 2.45±0.04 | 0.74±0.03 | 0.46±0.03 | 0.95±0.08 | 0.77±0.06 |
| | **GAN + Ours** | 70.04±2.97 | **9.77±0.22** | **2.62±0.05** | 0.81±0.01 | **0.47±0.01** | 1.28±0.04 | 0.87±0.01 |
| | **cGAN-PD + Ours** | **79.74±0.78** | 10.30±0.48 | 2.45±0.06 | **0.85±0.01** | 0.47±0.02 | 1.45±0.06 | **0.88±0.00** |

and are denoted as *oracle models*. In the experiments, S2M sampling is adopted on unconditional GAN and cGAN-PD. When adopting S2M sampling to cGAN-PD, a class containing the target joint class is used as the conditional value of cGAN-PD. BigGAN [2] is used as the backbone architecture for all generative models, and only single positive labels are used except for the oracle models.

To demonstrate the effectiveness of our S2M sampling method, we use three real-world image datasets: (i) **CIFAR-7to3** that has three classes ($A$, $B$, $C$), each of which contains four original classes of CIFAR-10, *i.e.* $A$={*Airplane*, *Automobile*, *Dog*, *Frog*}, $B$={*Automobile*, *Bird*, *Cat*, *Frog*}, $C$={*Cat*, *Deer*, *Dog*, *Frog*}, (ii) **CelebA-BMS** that consists of human face images, each of which is labeled for one of three attributes (*Black-hair*, *Male*, and *Smiling*), and (iii) **CelebA-ABMNSW** that consists of human face images, each of which is labeled for one of six attributes (*bAngs*, *Black-hair*, *Male*, *No-beard*, *Smiling*, and *Wearing-lipstick*). As depicted in Figure 4 (b-d), these datasets can have up to $2^3 - 1$ or $2^6 - 1$ joint classes. For S2M sampling, samples are obtained at 200 MCMC iterations for CIFAR-7to3 and CelebA-BMS, and obtained at 1000 MCMC iterations for CelebA-ABMNSW.

**Quantitative Results.** Table 2 summarizes the quantitative results of joint class generation on CIFAR-7to3, CelebA-BMS, and CelebA-ABMNSW. As expected, we can observe that cGAN-PD* and AC-GAN* cannot accurately generate samples from joint classes, which is consistent with the findings of Kaneko *et al.* [19]. Although CP-GAN shows reasonable accuracy for all experiments, it

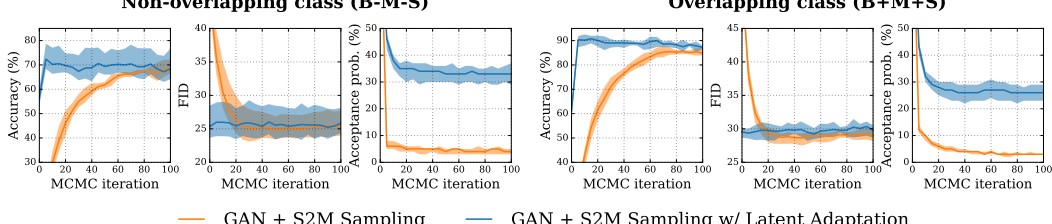

**Non-overlapping class (B-M-S)**  **Overlapping class (B+M+S)**

— GAN + S2M Sampling  — GAN + S2M Sampling w/ Latent Adaptation

Figure 7: Accuracy (%), FID, and acceptance probability (%) per MCMC iteration for S2M sampling with and without latent adaptation.

suffers from generating diverse samples. As discussed in Section 1, the high FID and low coverage support that CP-GAN tends to generate samples in a narrow scope of real data space.

In all experiments, our S2M sampling method adopted to cGAN-PD consistently outperformes the non-oracle baselines in terms of accuracy and diversity, *e.g.*, in CIFAR-7to3, our method improves accuracy and FID of CP-GAN by $38\%$ and 13.74, respectively. Such improvements verify that our S2M sampling method generates correct samples for joint classes, and also preserves diversity within them. Despite a large sample space of GAN, S2M sampling adopted to unconditional GAN also surpasses non-oracle baselines in terms of both fidelity and diversity while achieving reasonable accuracy. Surprisingly, despite the fact that the oracle models are trained with fully annotated data, S2M sampling shows a comparable performance against those models using only single positive labels. Moreover, as an post-processing method, S2M sampling fully preserves the image quality of GANs and shows the highest fidelity in terms of precision and density.

**Qualitative Results.** Figure 6 depicts the results of S2M sampling for every possible joint classes on CIFAR-7to3 and CelebA-BMS. The results show that S2M sampling can draw diverse high-quality images from the distributions of all joint classes. Furthermore, S2M sampling is model-agnostic and does not require the retraining of GANs; thus, it can be easily integrated with various GANs including state-of-the-art GANs. We depict more qualitative results of adopting our S2M sampling method to unconditioanl GAN and pretrained StyleGANv2 in Appendix E.5 and Appendix F.

**Analysis on Latent Adaptation.** To demonstrate the effect of latent adaptation, we tested our latent adaptation technique in a overlapping class (B+M+S) and a non-overlapping class (B-M-S) on the CelebA-BMS dataset. We first perform S2M sampling on unconditional BigGAN in these classes and draw $10k$ pilot latent samples obtained at 100 MCMC iterations. Using these latent samples, a Gaussian mixture of eight components is fit as a new latent of GAN (See Appendix B.2). Due to the discrepancy between the generator distribution and the target distribution, S2M sampling originally shows a low acceptance probability, as shown in Figure 7. In this case, applying latent adaptation increases the acceptance probability and significantly improves both accuracy and FID at the early stage of iterations. This indicates that the generator distribution becomes closer to the target distribution by adopting latent adaptation, resulting in an accelerated convergence speed.

## 5   Limitations

In general, as the number of attributes increases, the number of each joint class samples in the training dataset are drastically reduced. This may cause two primary problems: (i) training GANs to be able to generate each joint class samples becomes challenging, and S2M sampling cannot draw samples in which GANs do not generate, and (ii) the computation time of the sampling procedure increases. In this case, it may be necessary to apply the latent adaptation technique to shorten MCMC chains. In principle, we can reduce the sampling time complexity to be proportional to the number of attributes (See Appendix C).

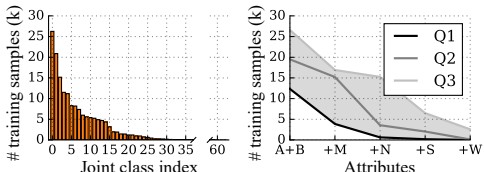

Figure 8: (Left) Distribution of the number of training samples for each joint class on CelebA-ABMNSW. (Right) As attributes are added, the quartiles of the number of samples for each joint class significantly decrease.

As shown in Figure 8, the number of training samples in joint classes rapidly decreases. Because of this, six attributes at most are used on the CelebA dataset for experiments. Almost half of the joint classes have fewer than 100 training samples when using six attributes (*i.e.*, CelebA-ABMNSW). Nevertheless, our method handles a larger number of attributes compared to the existing methods. We believe that our work can be further extended to larger attributes with the sufficient amount of data.

## 6 Conclusion

In this study, we investigate the single positive setting in the class-conditional generation task and propose a novel sampling framework called S2M sampling. We demonstrate that our proposed S2M sampling method can accurately draw high quality multi-label samples in the single positive setting. To improve sampling efficiency, we also introduce the latent adaptation technique. We believe that our theoretical framework provides a better understanding for weakly-supervised multi-label data generation. Our S2M sampling method has the potential to be applied to a wide variety of generative models as well, including clustering based GANs [42] and diffusion models [43, 44], among others.

## Acknowledgments and Disclosure of Funding

This work was supported by the Institute of Information & communications Technology Planning & Evaluation (IITP) grant funded by the Korean government (MSIT) (No. 2019-0-00075, Artificial Intelligence Graduate School Program (KAIST) and No. 2021-0-01778, Development of human image synthesis and discrimination technology below the perceptual threshold). This work was also supported by the National Research Foundation of Korea (NRF) grant funded by the Korean government (MSIT) (No. NRF-2022R1A2B5B02001913).

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
