# OpenReview forum: "Mining Multi-Label Samples from Single Positive Labels"
_NeurIPS.cc/2022/Conference — NeurIPS 2022 Accept_

### Official Review · Reviewer_wf99 · 2022-06-21

**Rating:** 5
**Confidence:** 3
**Soundness:** 2 fair
**Presentation:** 3 good
**Contribution:** 2 fair

**Summary:**

This paper proposes a sampling method based on MCMC called single to multiple sampling (S2M). A few applications for this “add-on” method on unconditional and conditional GAN is applied to show the effectiveness of the proposed method.

**Questions:**

The main questions for this paper are :

(1) Lack of sufficient comparison with other methods in this area in both theoretical study and experimental analysis (2) the motivation is not clear (especially for low data problem) (3) the insufficient experimental validation

**Limitations:**

N.A.

**Strengths And Weaknesses:**

Strengths:

The proposed method seems to be able to help resolving the multiple label learning from single positive samples to some extent, which may serves as an interesting supplement to the existing methods, although it is not clear how good it is compared to other methods.

Weakness:
(1) One of the main concerns for this paper is that on the topic of multiple label learning from single positive samples, there are already quite a few solid study from previous work. For instance, in the reference [1]:
Cole et al “Multi-Label Learning from Single Positive Labels”, CVPR 2021.
The submission fails to connect their work to the reference [1]. Thus, it is not clear the importance of the S2M method for improving this type of problem. For example, how is the performance comparison of the proposed method vs the techniques described in [1] including “label smoothing, expected positive regularization, pseudo-negative sampling, etc” in both theoretical analysis and experimental study.
Without such important study, the novelty and significance of the paper does not stay on the safe ground.

(2) It is also not clear how does the proposed method scale to large dataset.

(3) While the paper proposes an interesting application to enhance the results with S2M sampling, the overall depth of the paper clearly falls below the Neurips acceptance bar. It will be nice that the author can motivate their method better. Does the S2M method provide any advantages compared to label smoothing, pseudo-negative sampling etc ?
(4)  As we know, the sampling based method usually needs to accumulate sufficient number of samples to be effective, however, typically in the regime of multi-label learning from single positive labels, the data/labels are limited. Thus it is questionable that practically, how useful the method can be.
(5) The experimental validation is not sufficient and convincing as it only considers a special case, which makes the application domain quite limited

---

> ### Author Response · Authors · 2022-08-02
> **Response to Reviewer wf99 [2/2]**
>
> **[C5 / Q3] Validity of the experiments.**
>
> Our S2M sampling aims to generate images conditioned by multiple attributes when only one attribute is specified for each image in training data of GANs. To demonstrate the effectiveness of our method, we mainly examine our method on six datasets: MNIST-3/5, MNIST-Even, FMNIST-Even, CIFAR-7to3, CelebA-BMS, and CelebA-ABMNSW.
>
> In MNIST-3/5, MNIST-Even and FMNIST-Even, one class is fully contained into another class (A $\subset$ B). With these datasets, we verify that our S2M sampling can generate the images of class (B - A). For instance, MNIST-Even has two classes: one for odd digits (A = {1, 3, 5, 7, 9}) and another one for all digits (B = {0, 1, 2, 3, 4, 5, 6, 7, 8, 9}). In this circumstance, our S2M sampling can generate the images of B - A = {0, 2, 4, 6, 8}. In Figure 5, we showed that our S2M sampling successfully generates only the images of even digits and shows superior results compared to other generative methods.
>
> We also examined our S2M sampling in situations where classes are overlapped. In CIFAR-7to3 and CelebA-BMS, three classes are overlapped. When three classes A, B, and C are overlapped, we can generate images from a variety of joint classes such as (A + B + C, A + B - C, A - B - C, …). In CelebA-BMS, only one of the attributes (Black-hair, Male, Smiling) is annotated for each image. Then we examine that our S2M sampling can generate images of joint classes, e.g., Smiling black-haired man (B + M + S), Smiling black-haired woman (B + S - M), etc.. as shown in Figure 6. Moreover, to explore the harder setting, we conduct experiments with the dataset CelebA-ABMNSW which contains six attributes: Bangs, Black-hair, Male, No-beard, Smiling, Wearing-lipstick. All data in this dataset is annotated with only one of these attributes, and at most 63 joint classes can be considered.
>
> In our experiments conducted with these datasets, we can observe that our S2M sampling shows superior results compared to other baselines for generating images from joint classes even when only one class is specified for each image in train data, in terms of both quantitatively and qualitatively. Furthermore, we additionally conduct the experiments with StyleGANv2 for generating high-resolution images and show that our latent adaptation technique can further improve the speed of generation.
>
> Consequently, we believe that our experimental setting can prove the effectiveness of our S2M sampling and properly tackles the conditional generation task with the minimal supervision. We also want to emphasize that the conditional generation task with limited supervision is an under-explored task due to its difficulty. To the best of our knowledge, our attempt is the first to handle six attributes at once and introduce complex architecture like SytelGANv2 to generate images of joint classes.
>
> If you have any concerns or recommendations for future experimental settings, please let us know.
>
> ---
>
> [1] Elĳah Cole, et al. "Multi-Label Learning From Single Positive Labels." CVPR. 2021.\
> [2] Ming Hou, et al. "Generative Adversarial Positive-Unlabelled Learning." ĲCAI. 2018.\
> [3] Siddarth Asokan, et al. "Teaching a GAN What Not to Learn." NIPS. 2020.\
> [4] Takuhiro Kaneko, et al. "Class-Distinct and Class-Mutual Image Generation with GANs." BMVC. 2019.

---

> > ### Comment · Reviewer_wf99 · 2022-08-04
> > **Response to Author**
> >
> > Thanks for the response especially on the clarification with multiple label learning problem and my concern has been mostly addressed.
> >
> > I have updated my score.

---

> ### Author Response · Authors · 2022-08-02
> **Response to Reviewer wf99 [1/2]**
>
> **[C1 / C3 / Q1] Comparison with baselines of multi-label learning (classification).**
>
> First of all, we highlight that the purpose of our method is to perform the “conditional generation” task, which is fundamentally different from the “multi-label learning (classification)” task.
>
> The methods mentioned by the reviewer label smoothing, expected positive regularization, and pseudo-negative sampling are introduced to perform ‘multi-label classification’ with the limited annotations [1]. Those methods mostly focus on the problem of how to make a classifier not too highly confident for unknown labels. For instance, pseudo-negative sampling is proposed to mitigate the impact of negative labels in binary cross entropy (BCE) loss.
>
> In contrast, we tackle the problem of generating samples from the joint classes when only a single attribute is specified for every image of the training dataset. For example, if we have two image sets: the images of ‘man’ class and the images of ‘black-hair’ class, our aim is to find a way to generate infinitely many images of ‘black-haired man’, ‘black-haired woman’, or ‘non black-haired man’ classes. To do this, we draw samples from the conditional density of the joint class using the Markov chain Monte Carlo (MCMC) method. We employ several classifiers to estimate the density ratios among real and generated samples and utilize it to compute the acceptance probability used to run MCMC method. As highlighted in our paper, the existing conditional GANs cannot properly generate joint class samples in the single positive setting. We use the term ‘multi-label’ to represent that our method enables users to specify all attributes of samples as multi-label data in conditional generation tasks even when only single positive labels are given in the training set.
>
> Due to the difference in tasks, a direct comparison with techniques for mentioned methods such as label smoothing, expected positive regularization, and pseudo-negative sampling is not possible. To demonstrate the effectiveness of our S2M sampling, we mainly compare our method with the existing weakly-supervised generative methods such as GenPU [2], RumiGAN [3], and CP-GAN [4].
>
> ---
>
> **[C2] S2M sampling with larger dataset.**
>
> To generate images of joint classes, our S2M sampling used three classifiers $D_v^\ast, D_r^\ast, D_f^\ast$ to compute the acceptance probability of MCMC method. Therefore, we can freely adopt our S2M sampling to generative models trained with larger datasets as long as we can train these classifiers. For our main experiments, we used the CelebA dataset which contains 202K face images and 40 facial attributes.
>
> ---
>
> **[C4 / Q2] S2M sampling with limited data.**
>
> We focus on conditional generation tasks and use the generator as a sampler for the sampling method. To properly generate the samples for the target joint class using S2M sampling, the generator used for our method should be trained with a sufficient amount of data of that joint class. That is, if the target class samples exist in a small proportion in the training data of GANs, the efficiency of the sampling algorithm decreases.
>
> To tackle this issue, we additionally propose the latent adaptation technique, which restricts the latent prior of GANs to generate the samples of joint classes with a high probability. In our experiments, we observe that our latent adaptation technique performs successfully for various joint classes, further improving the practicality of our S2M sampling. Specifically, we provide an application of the latent adaptation technique to reduce the time complexity of the sampling algorithm proportional to the number of attributes in Appendix C.
>
> ---
>
> **[Q2] Motivations of our S2M sampling.**
>
> We want to remark that the main motivation of S2M sampling is generating images from joint classes (e.g., Black-haired man, Black-haired woman, Non-black haired man) when only one attribute (e.g., Black-hair, Male) is specified for each image in train data.
>
> To clarify this motivation, we first checked whether the existing generative model such as conditional GANs (cGANs) and CP-GAN can generate samples of joint classes on the synthetic dataset. Specifically, we conduct two dataset: (i) 1D Gaussian examples consisting of two classes with overlapped regions (See Figure 2) and (ii)  2 x 16 Gaussian examples consisting of grids of two classes with overlapped regions (See Figure 11 in Appendix).
>
> On these datasets, we confirmed that the existing methods cannot properly generate the samples of joint classes. Subsequently, we show that the density of joint classes can be computed using the conditional densities of single positive labels. We suggest using a MCMC method to draw samples from our proposed density, and validate the effectiveness of our method in both datasets.

---

### Official Review · Reviewer_UrwN · 2022-07-10

**Rating:** 6
**Confidence:** 2
**Soundness:** 3 good
**Presentation:** 3 good
**Contribution:** 3 good

**Summary:**

This paper proposes a method to generate multi-label images by using a (conditional-)GAN architecture. Because it is quite expensive to annotate a large dataset with multi-label annotations, this paper focuses on the single positive setting where each image is annotated with a single positive label. The paper proposes a novel sampling framework based on the Markov chain Monte Carlo method for generating multi-label images from single positive labels. The proposed method shows good performances in two settings: sampling for classes with inclusion and sampling for multiple classes.

**Questions:**

- I wonder what was the motivation to focus only on the single positive setting instead of a more generic setting e.g. at least a single positive label. For example, OpenImage dataset has a lot of partially-labeled images.

**Limitations:**

I think the authors have adequately addressed the limitations and potential negative societal impact of their work

**Strengths And Weaknesses:**

**Strengths:**

+ I think the idea of generating multi-label images is interesting. As mentioned in the paper, building large scale multi-label datasets is expensive so the approach of using partially labeled images looks very promising. Each image is annotated with a single positive label.
+ The problem formulation with intersection index set and difference index set seems novel.
+ The paper proposes a novel sampling framework based on the Markov chain Monte Carlo method for generating multi-label data from single positive labels.
+ The proposed method is evaluated in two settings:
  - sampling for classes with inclusion, where one class is entirely contained in another class. The proposed method shows better results on MNIST-3/5, MNIST-Even, and FMNIST-Even than RumiLSGAN and CP-GAN.
  - sampling for multiple classes, where multiple classes are overlapping. The proposed method shows better results on CIFAR-7to3, CelebA-BMS, and CelebA-ABMNSW than existing models like CP-GAN.

  For both settings, some visual results are shown and multiple metrics are used to evaluate the model performances.

+ The description of the limitations is interesting because it gives some insights about some potential next steps to improve the proposed method.

**Weaknesses:**

- The proposed approach seems to be specific to GAN, and cannot be used directly in other generative models like VAE.
- There is no information on how the examples in ​​Figure 5 and 6 are chosen.

---

> ### Author Response · Authors · 2022-08-02
> **Response to Reviewer UrwN**
>
> **[C1] The applicability of our S2M sampling for other generative models like VAE.**
>
> Since all of the related sampling schemes [1, 2, 3, 4] and baseline generative models [5, 6, 7] are GAN-based approaches, we mainly adopt our S2M sampling to GANs and evaluate the generation performance in the setting of GANs. However, in fact, our S2M sampling can be adapted to any generative model such as VAE, flow-based generative model because we do not have any assumption on the generator except that $supp p_t \subseteq supp p_G$ in our derivation. To adopt our S2M sampling to other generative models, only things we need to are the classifiers $D_v^\ast, D_r^\ast, D_f^\ast$ which are used to compute the density ratios used in the Metropolis-hastings algorithm. These classifiers can be trained using generated samples and real data with single positive labels. As you suggested, introducing our S2M sampling for various types of generative models will be an interesting direction for future work, and we will add formal discussion to our Appendix.
>
> ---
>
> **[C2] How are the samples in qualitative results (Figure 5 and 6) selected?**
>
> Thank you for pointing out the issue. In Figure 5 and Figure 6, we depict the qualitative results of our S2M sampling for each joint class appearing in four datasets: MNIST-Even, FMNIST-Even, CIFAR-7to3 and CelebA-BMS. In our paper, all samples for qualitative results are randomly selected from the outputs of our S2M sampling. We use the same model for quantitative results and qualitative results and do not use the separately trained models for depicting image samples. Following your comment, we will add the formal discussion for the experimental settings in the revised version.
>
> ---
>
> **[Q1] Motivation to focus on the single positive setting, rather than partially-labeled datasets?**
>
> As you mentioned, the partially-labeled datasets can be considered as the generalized setting for tackling the lack of limited annotations. The single positive setting, i.e., only a single positive label is observed for each image, is the special case of partially-labeled datasets with minimal supervision. We believe that handling the class-conditional generation task in the single positive setting is worthwhile due to following three reasons:
>
> * In the real world, collecting images for a single attribute is much easier than collecting fully annotated images. The single positive setting allows users to significantly reduce annotation costs for the multi-label data generation.
> * We can readily convert any partially-labeled datasets into datasets with the single positive labels by removing other labels except for a single positive label or relabeling a group of known labels as a single positive label.
> * Introducing the single positive setting can provide the insights for minimal supervision to generate multi-label data in the class-conditional generation task.
>
> As mentioned above, any partially-labeled datasets can be converted into datasets with single positive labels. However, in that case, there will be a loss of information. So, the generalized version of our S2M sampling which can cooperate with any partially-labeled datasets will be a prominent direction for future work. Thank you for your constructive suggestions!
>
> ---
>
> [1] Samaneh Azadi, et al. "Discriminator Rejection Sampling." ICLR. 2019.\
> [2] Ryan D. Turner, et al. "Metropolis-Hastings Generative Adversarial Networks." ICML. 2019.\
> [3] Tong Che, et al. "Your GAN is Secretly an Energy-based Model and You Should Use Discriminator Driven Latent Sampling." NIPS. 2020.\
> [4] Yifei Wang, et al. "Reparameterized Sampling for Generative Adversarial Networks." Machine Learning and Knowledge Discovery in Databases. Research Track - European Conference. 2021.\
> [5] Ming Hou, et al. "Generative Adversarial Positive-Unlabelled Learning." ĲCAI. 2018.\
> [6] Siddarth Asokan, et al. "Teaching a GAN What Not to Learn." NIPS. 2020.\
> [7] Takuhiro Kaneko, et al. "Class-Distinct and Class-Mutual Image Generation with GANs." BMVC. 2019.

---

### Official Review · Reviewer_NQQq · 2022-07-10

**Rating:** 7
**Confidence:** 4
**Soundness:** 4 excellent
**Presentation:** 3 good
**Contribution:** 3 good

**Summary:**

This paper proposed a sampling method based on MCMC to sample multi-label data from GANs trained with single positive labels. The authors prove that the target distribution can be derived from the conditional densities of single positive labels. Then they use several classifiers to estimate the density ratio and Metropolis-Hastings, an MCMC-based sampling method to draw samples. The method is technically sound and achieves good restuls in empirical studies.

**Questions:**

Why did the authors choose Metropolis-Hastings rather than other MCMC methods? Is there any reason or advantage?

**Limitations:**

The authors adequately addressed the limitations in Section 5 and the potential negative societal impact in Appendix F.

**Strengths And Weaknesses:**

Strength
1. The problem is interesting and valuable for reducing annotation costs.
2. The sampling method is novel and can be applied to any existing GANs.
3. The formulations and derivations are clear.
4. The empirical studies, including experiments on 1-D Gaussian mixtures and real image datasets, well support the method's superiority with a clear margin.

Weakness
1. As discussed in Limitations, the method is hard to scale up for many attributes. The authors may consider moving some conclusions of complexity analysis from the appendix to the main paper.
2. The experiment setting of sampling for multiple classes could be clearer with some examples. For instance, in CIFAR-7to3, class A includes images of airplanes, automobiles, dogs, and frogs.

---

> ### Author Response · Authors · 2022-08-02
> **Response to Reviewer NQQq**
>
> **[C1] Moving the conclusion of complexity analysis to the main paper.**
>
> The appendix contains the application of the latent adaptation technique which reduces the complexity of the sampling algorithm to be proportional to the number of attributes. As you suggested, we add a brief conclusion of this complexity analysis to the main paper in the limitation Section.
>
> ---
>
> **[C2] The detailed description of the experimental settings (e.g., CIFAR-7to3).**
>
> We appreciate the constructive comments! We clarify three classes A, B, and C defined in CIFAR-7to3 and describe how each joint class of CIFAR-7to3 corresponds to one original class of CIFAR10 in Section 4.2.
>
> ---
>
> **[Q1] Why do we use the Metropolis-Hastings algorithm rather than other MCMC methods?**
>
> As you mentioned, the different types of sampling methods can be used to tackle our problem. At the beginning of our study, we mainly considered applying three sampling algorithms which can be found in previous GAN sampling studies [1,2,3,4]; Rejection sampling, Independent Metropolis-Hastings algorithm, and Langevin dynamics. Here, we briefly discuss the pros and cons of each sampling method we faced in our problem settings.
>
> **(i) Rejection Sampling:**
> As discussed in DRS [1], the rejection sampling can be applied to sample from the target distribution $p_t(x)$ if we can compute the ratio between the target density $p_t(x)$ and the generator density $p_G(x)$, and the upper bound of the density ratio $p_t(x) / p_G(x)$. However, it is in general difficult and expensive to compute this upper bound in the high dimensional data space. We need several heuristics to mitigate the issues, e.g., shifting the logit score of the classifier used to compute the density ratio as introduced by DRS [1], which may introduce additional non-trivial efforts for hyperparameter searching.
>
> **(ii) Independent Metropolis-Hastings algorithm:**
> This algorithm can also be used to draw samples from the target distribution if we can compute the density ratio $p_t(x) / p_G(x)$. Unlike Rejection sampling, we do not need to compute the upper bound of this density ratio, but a sequence of samples forming a Markov chain is required. Our study mainly used this algorithm because it empirically performed well without complex heuristics and the sampling accuracy can be readily controlled at the cost of MCMC steps. To further mitigate the sample efficiency in our problem, we suggest the latent adaptation technique.
>
> **(iii) Langevin dynamics:**
> Langevin dynamics is a gradient-based MCMC approach which can also be used when we can compute the density ratio $p_t(x) / p_G(x)$. Several studies~[3, 4] employ its Euler-Maruyama discretization to improve the quality of GAN samples. While this algorithm can efficiently push a chain of samples towards the target distribution, the step size of the algorithm is very sensitive to the sampling cost and quality. Especially, in our problem, we need to deal with the case that the sample falls within the space where the gradient is not well-defined, i.e. $supp p_G \setminus supp p_t$. We did not use the algorithm as we could not find an effective way to address these issues.
>
> In addition to the algorithms listed above, we expect that there may be efficient sampling algorithms that can effectively solve the problem we explored, which can be attempted as a future work. Since we expect our consideration can be helpful for the machine learning community to deal with similar problems, we add this discussion to the appendix G of our paper.
>
> ---
>
> [1] Samaneh Azadi, et al. "Discriminator Rejection Sampling." ICLR. 2019.\
> [2] Ryan D. Turner, et al. "Metropolis-Hastings Generative Adversarial Networks." ICML. 2019.\
> [3] Tong Che, et al. "Your GAN is Secretly an Energy-based Model and You Should Use Discriminator Driven Latent Sampling." NIPS. 2020.\
> [4] Yifei Wang, et al. "Reparameterized Sampling for Generative Adversarial Networks." Machine Learning and Knowledge Discovery in Databases. Research Track - European Conference. 2021.

---

> > ### Comment · Reviewer_NQQq · 2022-08-06
> > **Thank the authors for the response**
> >
> > My questions are well addressed and the revision looks good to me.
> > Thank the authors for the response.

---

### Official Review · Reviewer_E5Ue · 2022-07-11

**Rating:** 6
**Confidence:** 4
**Soundness:** 3 good
**Presentation:** 4 excellent
**Contribution:** 3 good

**Summary:**

This paper focus on a setting that multi-label with only one positive label and try to generate the multi-label data with generative adversarial network (GAN). The proposed method considers both unconditional and class conditional version by a pre-trained GAN in a sampling strategy. The experimental results shows that the proposed method gets a promising result under the single positive setting.

**Questions:**

Please see my main review.

**Limitations:**

The work can only apply to the small datasets or small class space datasets and the experiments need to be further improved.

**Strengths And Weaknesses:**

Strengths:

Overall, the problem of multi-label with single positive label is interesting.

The paper is well written and organized.

The idea of using single class densities to provide an estimate for the multi-class target distribution is interesting.

Weaknesses:

The datasets used in the experiments can be relatively small. It is better to adopt some large datasets to validate the effectiveness of the proposed methods.

Since the paper focuses on the multi-label with singe positive label setting, it is better to use some datasets with large label space to see the performance the proposed method.

I suggest the author to add an experiment about varying the label space of the data and show how the performance changes to further validate the multi-label data generation performance.

Overall, this paper is novel to some extent. It is better that more promising results from some large datasets.

---

> ### Author Response · Authors · 2022-08-02
> **Response to Reviewer E5Ue**
>
> **[C1] Experiments on large datasets.**
>
> Thank you for your suggestion. To demonstrate the effectiveness of our S2M sampling, we used the CelebA dataset for our main experiments. Original CelebA dataset contains 202K face images and each image has 40 facial attributes with full annotation. Currently, it is difficult to find larger multi-label image datasets with larger scales that are widely used in the generation tasks. The main goal of our study is to introduce the single positive setting in the generation tasks, and demonstrate that multi-label data can be generated using single positive labels. We believe that CelebA is a sufficient dataset to achieve our research goals and to show the effectiveness of S2M sampling compared to existing weakly–supervised generative baselines such as CP-GAN.
>
> Nevertheless, as the reviewer suggested, we also believe that our S2M sampling can be effectively used on larger datasets since our method still has certain limitations in generating for a joint class with a few samples in the training dataset. In further discussion, we will sincerely consider other multi-labeled datasets for generation if you can recommend a few of them.
>
> ---
>
> **[C2] Experiments on large label space.**
>
> As you noticed, theoretically, our S2M sampling can be used regardless of the number of attributes. For the experiments, we attempted to increase the number of the attributes, and conclude that at most six attributes can be performed for the evaluation on CelebA dataset as we stated in the limitation Section. For instance, only 1.95% of the real samples for joint class B+M+S-A-N-W exist in the CelebA training set, and this makes the generator used for our S2M sampling suffer from learning samples for the corresponding joint class. Basically, we need a lot of data that is exponentially proportional to the number of data attributes to properly generate samples for all joint classes.
>
> To our best knowledge, our method is the first attempt to handle the joint classes of six attributes in the generation task while the previous baselines GenPU[1], Rumi-GAN[2], and CP-GAN [3] attempted to handle two or three attributes. It is still challenging to generate multi-data by reflecting a large number of attributes in the single positive setting, and it would thus be interesting to explore this problem in future works.
>
> ---
>
> **[C3] Experiments with other combinations of attributes.**
>
> Following your suggestions, we conduct additional experiments with a choice of other attributes on the CelebA-HQ dataset with resolution of 256x256. We additionally consider two groups of attributes: {Brown-hair, Bushy-eyebrows, Mouth-slightly-opens} (CelebA-BBM) and {Bags-under-eyes, High-cheekbones, Wavy-hair} (CelebA-HBW). We chose those attributes since they have strong visual impact and can easily be detected in qualitative evaluations.
>
> For the experiment, we use pre-trained StyleGANv2 as the baseline, we draw samples for each joint class for two groups of attributes and compute the accuracy and FID. For quantitative evaluation, we measure the performance of S2M sampling adopted to StyleGANv2, compared to that of original StyleGANv2 which generates randomly-conditioned images. Along with the quantitative results, we report the qualitative results in Appendix F.
>
>
> \begin{array}{l | l cc}
> \hline
> \text{Dataset} & \text{Method} & \text{Accuracy}(\uparrow) & \text{FID}(\downarrow) \newline
> \hline
> \ CelebA-BBM & \text{StyleGANv2} & 18.41\\% & 15.57  \newline
> \              & \text{StyleGANv2 + Ours} & \mathbf{74.97\\%} & \mathbf{14.41}  \newline
> \hline
> \ CelebA-HBW & \text{StyleGANv2}& 12.68\\% & 15.80  \newline
> \              & \text{StyleGANv2 + Ours} & \mathbf{70.94\\%} & \mathbf{14.54}  \newline
> \hline
> \end{array}
>
> ---
>
> [1] Ming Hou, et al. "Generative Adversarial Positive-Unlabelled Learning." ĲCAI. 2018.\
> [2] Siddarth Asokan, et al. "Teaching a GAN What Not to Learn." NIPS. 2020.\
> [3] Takuhiro Kaneko, et al. "Class-Distinct and Class-Mutual Image Generation with GANs." BMVC. 2019.

---

> > ### Comment · Reviewer_E5Ue · 2022-08-08
> > **Thanks for the author's response**
> >
> > I have read the author's response. My questions are addressed to some extent.

---

### Author Response · Authors · 2022-08-02
**General Response**

Dear reviewers and AC,

We sincerely appreciate your efforts in reviewing our manuscripts: Mining Multi-Label Samples from Single Positive Labels.

We are grateful that reviewers found the problem we tackle is interesting (E5Ue, NQQq, UrwN) and valuable for reducing annotation costs (NQQq, UrwN), and the proposed S2M sampling is novel (NQQq, UrwN) and shows its effectiveness with the promising results in extensive experiments (E5Ue, NQQq, UrwN). As the reviewer mentioned, we believe that our proposed method provides meaningful insights in generating conditional images with limited supervision.

Within the short response period, we did our best to reflect all reviewers’ feedback. In addition to answering each reviewer’s questions, we also faithfully revised our manuscript following the suggestions. The revisions made during this period are highlighted in blue.

Sincerely,
Authors.

---

### Author Response · Authors · 2022-08-08
**Thank you for your thoughtful discussions**

Dear reviewers,


We greatly appreciate your efforts in reviewing our manuscript.

We hope that our responses and discussions have addressed the reviewers’ concerns.

In case of any further issues, we will do our best to address them as soon as possible.

Thank you very much again for your time and efforts in giving constructive reviews.


Sincerely,

Authors.

---

### Meta-Review · Area_Chair_aZx7 · 2022-08-27

**Recommendation:** Accept
**Confidence:** Certain

**Metareview:**

This paper addresses the GAN generation with multi-label condition variables problem by a single-to-multi-label generation method. More specifically, given only one positive label for each image, this paper proposes a MCMC sampling method to generate combinations of labels and reduce the cost for data annotation. This is a challenging problem and the authors have made clear assumptions under which the sampling can be successful. The experimental results demonstrate the effectiveness of the proposed method.


Overall, the paper is novel and interesting.  I would recommend acceptance of this paper given the novelty of the idea and the technical soundness. However, I would suggest that the authors could narrow the scope of “multi-label” to multiple attributes. It is unclear whether the proposed method can handle the general multi-label problem in which multiple objects could appear in an image.


**Award:**

No

---

### Decision · Program_Chairs · 2022-09-14

Accept